Biodegradation of polyethylene in digestive gland homogenates of marine invertebrates

Istomina Aleksandra 1
Chelomin Victor 1
http://orcid.org/0000-0002-4028-0567 Mazur Andrey 1 mazur.aa@poi.dvo.ru
Zhukovskaya Avianna 1
Karpenko Alexander 2
Mazur Marina 2
1 V.I. Il’ichev Pacific Oceanological Institute, Far Eastern Branch, Russian Academy of Sciences , Vladivostok , Russia
2 A.V. Zhirmunsky National Scientific Center of Marine Biology, Far Eastern Branch, Russian Academy of Sciences , Vladivostok , Russia
Zhukova Natalia
Electronic publication date: 2024 Feb 26
Publication date: 2024
Volume: 12
Electronic Location ID: e17041
Received 2023 Nov 20; Accepted 2024 Feb 12
Copyright: © 2024 Istomina et al.
Copyright year: 2024
Copyright holder: Istomina et al.
License: This is an open access article distributed under the terms of the Creative Commons Attribution License, which permits unrestricted use, distribution, reproduction and adaptation in any medium and for any purpose provided that it is properly attributed. For attribution, the original author(s), title, publication source (PeerJ) and either DOI or URL of the article must be cited.
License URL: https://creativecommons.org/licenses/by/4.0/

Keywords: Biodegradation of plastic, Fourier transform infrared spectroscopy

Funding: State assignment for research work of V.I. Il’ichev Pacific Oceanological Institute, FEB RAS 121021500052-9 "Popov Island" Il’ichev Pacific Oceanological Institute The research was supported by the State assignment for research work of V.I. Il’ichev Pacific Oceanological Institute, FEB RAS (no. 121021500052-9) at the marine experimental station “Popov Island” Il’ichev Pacific Oceanological Institute. The funders had no role in study design, data collection and analysis, decision to publish, or preparation of the manuscript.

==============================
Вiotic factors may be the driving force of plastic fragmentation along with abiotic factors. Since understanding the processes of biodegradation and biological depolymerization of plastic is important, a new methodological approach was proposed in this study to investigate the role of marine invertebrate digestive enzymes in plastic biodegradation. The aim of this study is to evaluate the possibility of enzymatic biodegradation of polyethylene fragments in the digestive gland homogenate of marine invertebrates differing in their feeding type (Strongylocentrotus nudus, Patiria pectinifera, Mizuhopecten yessoensis). Significant changes are found in the functional groups of the polymer after 3 days of incubation in the digestive gland homogenates of the studied marine invertebrates. A significant increase in the calculated CI (carbonyl index) and COI (сarbon-oxygen index) indices compared to the control sample was observed. The results suggest that digestive enzymes of studied organisms may play an important role in the biogeochemical cycling of plastic.

Introduction

The problem of plastic pollution in the environment is a pressing issue for which a solution is urgently needed. Solar radiation, temperature, mechanical abrasion, and other physicochemical processes contribute to the destruction of macro and mesoplastics to microplastics (MPs) (fragmentation) (Lambert & Wagner, 2016; Ward et al., 2019; Chamas et al., 2020; Mattsson et al., 2021; Born & Brüll, 2022; Duan et al., 2022; Xi et al., 2022). A multitude of studies shows the global distribution of microplastic particles, that can be found in all environments (water, land, air) (Koelmans et al., 2014; Avio et al., 2015; Li, Tse & Fok, 2016; Xia, Niu & Yu, 2023). Moreover, microplastics can be taken up by a variety of hydrobionts of different taxa and trophic levels, which has been reported by several studies (Bom & Sá, 2021; Dellisanti et al., 2023). The ingestion of microplastics can have harmful effects on biota (Revel et al., 2019; Trifuoggi et al., 2019; Thomas et al., 2020; Chelomin et al., 2022).

It is believed that biotic factors may also be a key driving force behind the plastic fragmentation process (Jang et al., 2018). For example, various organisms are able to shred plastic fragments to micro- and nanoplastics by passing them through their digestive system (Hodgson, Bréchon & Thompson, 2018; Dawson et al., 2018). It is shown that when passing through the digestive system of the Antarctic krill Euphausia superba, the plastic broke down into small fragments up to 1 µm in diameter (Dawson et al., 2018). When MPs passed through the digestive tract of the polychaete Marphysa sanguinea, hundreds of thousands of its particles were produced in 1 year (Jang et al., 2018). It was observed that ingestion and excretion of MPs by the sea cucumber Holothuria tubulosa increased their bioavailability, although they did not change the size. Labeled MPs from pseudofeces of H. tubulosa were resuspended in much greater numbers than MPs from bottom sediment (Bulleri et al., 2021). In addition, Rani-Borges et al. (2023) found a decrease in the diameter of PS particles up to 25.3% smaller than the initial size after interaction with the digestive tract of amphipod Hyalella azteca within 7 days. It is suggested that biological activity of marine organisms inhabiting plastic debris floating in the sea may accelerate its degradation and contribute to the generation of MPs along with abiotic factors (Jang et al., 2018). For example, microscopic images of rope surfaces (made of polypropylene, polyethylene, and nylon) reveal noticeable surface roughness and the decrease in mechanical properties and weight of the sample presumably due to the action of fouling organisms (macroalgal species, the periwinkle Littorina littorea, the amphipod Stenula sp., the barnacle Eliminus modestus, and the blue mussel, Mytilus edulis) (Welden & Cowie, 2017). In all these studies, the authors suggest that digestive enzymes of multicellular organisms also contribute to the fragmentation of plastic particles along with mechanical action.

Recently, bioremediation was considered as an alternative method to reduce the presence of microplastics in the environment (Miri et al., 2022). For bioremediation, bacteria were isolated from mangrove forests, waste disposal dumps and sewage treatment plants, and cow dung (Jadaun et al., 2022). In addition, bacterial communities from the surface of multicellular marine organisms belonging to the types Annelida, Cnidaria, Hydrozoa, Polifera and Tunicata were isolated and cultured (Villalobos, Costa & Marín-Beltrán, 2022). Among them, several genera of microorganisms colonizing the plastic were identified. Members of a diverse group of taxa including diatom algae, infusoria and bryozoans have similarly been found with plastic (Syranidou et al., 2019).

Marine invertebrates such as annelids (sand worms) and echinoderms (sea cucumbers) were utilized for bioremediation of microplastics in wastewater treatment plants (Miri et al., 2022). Mangrove rhizospheres were shown to act as a sink for MPs, and sediments of seagrasses such as Zostera marina and Enhalusa coroides can trap MPs (Miri et al., 2022). Microorganisms in the larval gut of lesser waxworm (Achroia grisella) play an important role in biodegradation and mineralization of polyethylene (Ali et al., 2023).

Plastic valorization and bioremediation (i.e., phytoextraction, composting, enzyme-mediated biodegradation) are promising strategies for controlling MPs pollution (Ru, Huo & Yang, 2020). A large variety of specific enzymes are known to be involved in various stages of polymer chain degradation (Lucas et al., 2008). In general terms, the activity of these enzymes ultimately leads to the oxidation and hydrolysis of polymers. Then it is followed by the fragmentation and release of low molecular-weight compounds (monomers and oligomers) that can be involved in metabolic processes. It should be emphasized that enzymes participate in the biodegradation processes of synthetic polymers. They are primarily catalysts focused on the digestion of natural polymers (cellulose, lignin, mucopolysaccharides) to extract energy for essential processes. On the basis of these assumptions, it is logical to assume that digestive enzymes of marine invertebrates involved in the digestion of cell walls of phytoplankton and algae, which include polymeric carbohydrates, are capable to oxidative hydrolysis of plastic.

Polyethylene is a typical petroleum-based plastic that is considered to be non-biodegradable due to its extreme strength. The resistance of polyethylene to microbiological degradation is due to its physicochemical features including its high molecular weight, polymer structure devoid of functional groups, and hydrophobicity (Somanathan et al., 2022; Ali et al., 2023).

Since understanding the biodegradation and biological depolymerization of plastic is important, this study proposes a novel methodological approach to investigate the role of marine invertebrate digestive enzyme complexes in plastic biodegradation. The aim of this study is to evaluate the feasibility of enzymatic biodegradation of polyethylene (PE) fragments in marine invertebrate digestive gland homogenate.

Materials and Methods

Site of animals collection and material

The mature animals were collected by divers in July 2023 in the waters of the Alekseev Bay in the Sea of Japan (42°59′N; 131°43′Е). Marine invertebrates differing in their feeding type were selected as objects of the study: black sea urchin (Strongylocentrotus nudus)—herbivores, starfish (Patiria pectinifera)—carnivores, and scallop (Mizuhopecten yessoensis)—filter feeder. In total, nine scallops, thirty black sea urchins, six starfish were used. Animals were immediately dissected, and the digestive glands were extracted. Tissue homogenate was prepared in 0.2M of phosphate buffer, pH 7.5 (2:1, w/v) (1 g of the digestive gland from one scallop, 2.7 g—from one starfish and 0.3 g—from one black sea urchin).

The plastic fragments (5 mm long and 5 mm wide each) used in the experiments were obtained by cutting unused polyethylene bag of the Thorog brand manufactured by CleanWrap Corp., Korea, the chemical composition of which was determined by IR spectroscopy (Fig. 1).

Figure 1 Original FTIR spectra of plastic bag used in experiments.

Experiment in vitro

Subsequently, the plastic fragments were incubated in 1.5 ml of tissue homogenate for 3 days (three times repeated for each species of animal), with daily replacement of the homogenate. When the homogenate was changed, the plastic fragments were washed in distilled water. Incubation was carried out with constant stirring (BIOSAN MultiBio RS-24) at room temperature 20 °С. After 3 days of incubation, the plastic fragments were washed in distilled water to remove the homogenate residues and then in 70% ethanol to remove the sorbed organic substances (proteins and fats) for 24 h under constant agitation. After washing, the plastic fragments were dried at room temperature for 1 day before analysis. Control plastic samples were incubated in the homogenization buffer and were subjected to the same washing and processing steps. Biodegradation of the plastic surface were assessed using FTIR spectroscopy.

Fourier-transform infrared spectroscopy

All images of FT-IR spectra were prepared using a spectrometer model IRAffinity-1S together with a total internal reflection attachment manufactured by Shimadzu (Japan). The following instrument settings were used to obtain spectra: wavelength range 4,000–400 cm−1, 32 scans per spectrum, spectral resolution 4 cm−1). For the measurement of the background value, we used the measurement by air with the above-described settings. Then the spectra were processed using the LabSolutions IR computer program manufactured by Shimadzu (Japan).

The degree of biodegradation of plastic fragments was evaluated using three different indices: carbonyl index (CI), hydroxyl index (HI) and сarbon-oxygen index (COI). CI is the most commonly used index to measure the chemical oxidation of polyolefins such as polyethylene and polypropylene and reflects the deterioration of the mechanical properties of these polymers (Rouillon et al., 2016). CI was determined using the SAUB (specified area under band) technique described in Almond et al. (2020). CI was calculated from the ratio between the integrated band absorbance of the carbonyl (C=O) peak from 1,850 to 1,650 cm−1 and that of the methylene (CH2) scissoring peak from 1,500 to 1,420 cm−1 as expressed in the following equation: CI = (Area under band 1,850–1,650 cm−1)/(Area under band 1,500–1,420 cm−1).

The measurement of peak area rather than intensity at a particular wavelength is based on the fact that the cleavage of polyolefins produces not only ketones at 1,714 cm−1, but also dozens of potentially different carbonyl products such as γ-lactones (1,780 cm−1), esters and/or aldehydes (1,733 cm−1), and carboxylic acids (1,700 cm−1). The area between 1,500 and 1,420 cm−1 was chosen as the reference range and remains distinct throughout the degradation process during photo- and thermo-oxidation (Almond et al., 2020).

Similarly, HI was calculated as the absorbance ratio of hydroxyl groups at 3,353–3,021 cm−1 and 1,504–1,467 cm−1 for reference peak (Campanale et al., 2023). COI was calculated as the ratio of the absorbance of carbon-oxygen groups at 924–1,197 cm−1 and the value of reference peaks at 2,987–2,866 cm−1 (Campanale et al., 2023).

Statistics

Statistical processing of the results was performed using Statistica 7. The Mann-Whitney’s U test for non-parametric variables was used to assess reliability of parameter changes. Significance was established at p < 0.001 and p < 0.01.

Results

The PE showed reference peaks displayed at 719 and 729 cm−1 (–CH2 rocking deformation), 2,846 and 2,915 cm−1 (–CH2 symmetric and asymmetric stretching), 1,462 and 1,471 cm−1(–CH=CH– stretching) (Fig. 1) (Campanale et al., 2023). Changes in the functional groups of the polymer occurred after 3 days of incubation in the homogenates of the digestive gland of marine invertebrates (Fig. 2).

Figure 2 FTIR spectra of PE after 3 days of incubation relative to control: (A) In the digestive gland homogenate P. pectinifera, (B) S. nudus, (C) M. yessoensis.

CI, сarbonyl index; HI, hydroxyl index; COI, сarbon-oxygen index.

The first evident difference in the spectra of the sample in contrast to the control one was the alteration of the area relative to the OH stretching (3,200 to 3,900 cm−1). This area indicates the appearance of hydroxyl groups (Ali et al., 2023). Besides a broad band of peaks was observed in the range 2,000–1,300 cm−1. The PE showed new absorption peaks in the carbonyl and double bond region centered at about 1,653 cm−1 (for P. pectinifera and S. nudus) corresponding to carboxylate (–COO–) and centered at 1,743 cm−1 (for M. yessoensis) corresponding to carbonyl (–C=O) groups. In turn, the peaks at 1,506, 1,521, 1,541, 1,558 cm−1 indicate an increase in the proportion of double bonds.

Only for PE sample incubated in scallop digestive gland homogenate, it is a increase in the absorption band in the range of 1,300–900 cm−1 (centered around 1,161 cm−1), corresponding to the spectral range of carbon-oxygen (carboxyl) bond (C–O).

The results of indices calculation are presented in Table 1. There is a significant increase in all calculated indices compared to the control sample, except for HI (S. nudus and M. yessoensis). The most pronounced change in CI was typical for PE incubated with homogenates of the digestive gland of the scallop (p = 0.0005)—to a lesser extent sea urchin (p = 0.003) and starfish (p = 0.003). The change in COI was pronounced for starfish (p = 0.002) and scallop (p = 0.0005) in contrast to sea urchin (p = 0.012).

Table 1 Comparison of the PE degradation indices after incubation in tissues homogenates.

CI, сarbonyl index; HI, hydroxyl index; COI, сarbon-oxygen index.

	СI	HI	COI	
Control	0	0	0.06 ± 0.05	
P. pectinifera	0.025 ± 0.01*	0.015 ± 0.022*	0.25 ± 0.08*	
	(p = 0.003)	(p = 0.019)	(p = 0.002)	
S. nudus	0.16 ± 0.02*	0	0.02 ± 0.02*	
	(p = 0.003)		(p = 0.012)	
M. yessoensis	0.79 ± 0,16*	0	0.19 ± 0.07*	
	(p = 0.0005)		(p = 0.0005)	
Notes:

Data is presented as mean value ± standard deviation for ten measuring.

* Significant differences vs. control (p < 0.001 and p < 0.01, Mann-Whitney U test).

Discussion

The results of our experiments showed that after PE exposure to homogenates of all studied marine organisms, oxygen-containing functional groups appeared in the PE structure, which to some extent indicates the activation of the initial stage of the polymer degradation process. According to modern concepts, biodegradation is a complex physico-chemical transformation of a polymer into smaller molecules involving living organisms. Conventionally, this process includes four consecutive interrelated stages: biodegradation, biofragmentation, bioassimilation and biomineralization. Polymer degradation is determined by the flexibility or mobility of the molecular chain, which is enhanced by the introduction of additional functional groups, especially oxygen-containing ones. Such oxidative degradation leads to changes in polymer chains, including a decrease in hydrophobicity, an increase in inter-chain volume and bioavailability for enzymes, changes in crystallinity, etc. In general, this contributes to the acceleration of depolymerization and degradation of the polymer (Jadaun et al., 2022; Miri et al., 2022). It is noteworthy that, in comparison with the experiments on the starfish, these processes were expressed to a greater extent when using homogenates of the digestive gland of the scallop and sea urchin. These differences are probably related to the nature of the diet of these animals. Even though the M. yessoensis is a filter feeder and the S. nudus is a herbivores, the diet of both animals is dominated by plant food (algal remains, microalgae), whereas the P. pectinifera is a carnivores, preferring food of animal origin.

Polysaccharides are natural polymers and are the main components of algae that are utilized as food by herbivorous marine invertebrates. Accordingly, polysaccharide degrading enzymes such as alginate lyase, mannanase, cellulose and laminarinase, have been found in the digestive tract of some mollusks (Milke, Bricelj & Ross, 2012; Lyu et al., 2016). Enzymes involved in polysaccharide metabolism have been identified within the lysosomal enzymes of the sea scallop Chlamys farrery (Lyu et al., 2016). There is a suggestion that herbivorous organisms may participate in the biodegradation of polystyrene due to the high activity of cellulase enzyme (Song et al., 2020; Yang et al., 2021). In addition, microplastic can be retained for a long time (6 days) in the digestive system of the mussel Mytilus galloprovincialis (Fernandez & Albentosa, 2019) and influence the activity of digestive enzymes (Trestrail et al., 2021).

It has been shown in the sea urchin that PET microplastic particles can fragment not only mechanically but also chemically when passing through the digestive tract of the animal (Parolini et al., 2020). According to the authors, either sea urchin digestive enzymes or microorganisms associated with the digestive system may be involved in this process. The authors attribute the weak changes in IR spectra to the short residence time of PET in the sea urchin digestive system (less than 24 h).

Although polyethylene biodegradation is a very slow process (Alshehrei, 2017), in our study we observed significant chemical changes in the surface of plastic fragments upon relatively short-term exposure to digestive gland homogenates from three representatives of marine invertebrates, which contain a complex of digestive enzymes.

Similar modifications of the IR spectra of PE microplastics were found at incubation with the fungus Zalerion maritimum (from 7 to 28 days) (Paço et al., 2017). A gradual increase in the intensity of the bands for the peaks at 3,700–3,000 cm−1 caused by hydroperoxide and hydroxyl groups was observed. Similarly, the areas at 1,700–1,500 and 1,200–950 cm−1 are due to carbonyl groups and double bonds, respectively. This could be the result of various oxidation reactions such as functional groups present in the polymer, leading to the observed increase in signal.

Moreover, similar results were obtained when PE was incubated with bacteria from the gut of lesser waxworm larvae A. grisella (30 days) (Ali et al., 2023). When compared with the control samples, the bacteria-treated film showed the appearance of several different peaks that are associated with the formation of –OH groups (3,367–3,600, centered at 3,404 cm−1). The peaks at 1,708 and 1,635 cm−1 were associated with the presence of carbonyl (–C=O) and carboxylate (–COO–) groups, respectively. It was suggested that the formation of carbonyl groups may result from bacterial adhesion and/or biodegradation of the polymer (Villalobos, Costa & Marín-Beltrán, 2022).

Changes on the functional groups on the surface of the PE plastic films have been demonstrated after their exposure to tailored marine consortia (Syranidou et al., 2019). As a result of microbial activity, new bands at 1,610, 1,560 and at 990 cm−1 appeared on the surface of the microbially treated PE films which correspond to C=C bond (Syranidou et al., 2019). The C=C bond formation may occur mainly because of the breakdown of the main chain carbon bond with a substituent (–C–R) and breakdown of C–H bond (De Paoli, 2008; De Bomfim et al., 2019).

Conclusion

Enzymes of the digestive system of studied marine invertebrates dramatically enhance the structural modification of PE as the initial stage of the biodegradation process. The infrared spectroscopy data clearly indicate changes in the structure of the polymer surface, i.e., the introduction of functional groups into the polymer chain, such as (C=O, C–O–R, C–OH, and C=C). Given that communities of specific symbiotic microorganisms are present in the digestive tract of marine invertebrates, the possibility remains that their enzymes are involved in plastic biodegradation. Further studies in this direction should be aimed at establishing the role of symbiotic microorganisms in plastic biodegradation.

Supplemental Information

Supplemental Information 1 Peak intensity values of IR spectra.

Additional Information and Declarations

Competing Interests

Author Contributions

Data Availability

The authors declare that they have no competing interests.

Aleksandra Istomina conceived and designed the experiments, performed the experiments, prepared figures and/or tables, and approved the final draft.

Victor Chelomin conceived and designed the experiments, analyzed the data, authored or reviewed drafts of the article, and approved the final draft.

Andrey Mazur performed the experiments, prepared figures and/or tables, and approved the final draft.

Avianna Zhukovskaya analyzed the data, authored or reviewed drafts of the article, and approved the final draft.

Alexander Karpenko analyzed the data, authored or reviewed drafts of the article, and approved the final draft.

Marina Mazur analyzed the data, prepared figures and/or tables, and approved the final draft.

The following information was supplied regarding data availability:

The FTIR spectra is available in the Supplemental File.

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
