# Peer review of "Biodegradation of polyethylene in digestive gland homogenates of marine invertebrates"

_PeerJ, doi:10.7717/peerj.17041_

## Round 0.1 · original submission · Major Revisions

Dear Dr. Mazur,

Your manuscript has been evaluated. The reviewers found the topic of the study is very interesting and fits well into the current research in the field of polymer degradation. However, the manuscript needs serious revision, especially taking into account the comments that the introduction should be expanded, the results and discussion should be detailed, and the conclusion should be based on the results of your research.

The reviewers provided detailed comments, and I ask that you consider these carefully when revising the manuscript.

When submitting the revised version of your manuscript, please state in your cover letter point-by-point which changes you have made in response to the reviews and where and why you have refused to follow a particular suggestion. The accordance between the changes and the reviewers' requests should be sufficiently transparent.

Reviewer 1 ·

Basic reporting

no comment

Experimental design

no comment

Validity of the findings

no comment

Additional comments

PeerJ
Biodegradation of polyethylene in digestive gland homogenates of marine invertebrates Manuscript Number: #93008


Dear authors, I have had the opportunity to review your manuscript titled " Biodegradation of
polyethylene in digestive gland homogenates of marine invertebrates" and overall, I find the content to be valuable and well-presented. I consider the results regarding the biodegradation by digestive enzymes of invertebrate organisms are highly relevant.

I would like to bring to your attention a few suggestions that can improve the final document.

I greatly appreciated as a reader the time and care the authors took to write this text.

I also would like to take this opportunity to suggest reading the following article, as I believe it may be of interest to the authors: https://doi.org/10.1016/j.aquatox.2023.106516

Please find below my suggestions and inquiries:

Line 40: The phrase sounds a little strange, perhaps a little too informal. I suggest some modification, as follows: The environmental challenge posed by plastic pollution is critical, demanding an urgent resolution.

Line 52: holothurian is written in lower case.

Lines 84-85: The sentence with the objectives would be better if it were linked with the two last paragraphs and not isolated.

Line 94: Fundamental information is missing from the text. How were the organisms collected? How many organisms were collected from each species? How were the organisms sorted? Or maybe there was no screening? Were the plastic bags the same color and polymer-type? Were they used or new? Obtained directly from the industry or purchased in stores? You can also add the morphology of your plastic sample.

Lines 101-103: For better understanding for the reader, I suggest the following change to the text: “Subsequently, the plastic fragments were incubated in 1.5 ml of tissue homogenate for 3 days, with daily replacement of the homogenate.”

Line 102: Why only 3 days? Why not do a test with a longer incubation period?

Line 103: Did you use filtered distilled water? If not, how can you guarantee that there has been no increase of plastic particles in your samples from the water?

Line 104: What was the room temperature of the lab?


Line 145/Figure 1: Is there any reason why the x-axis of the graphs is not continuous?
I think it would be more interesting to present the overlapping of the original plastic spectra and those after a 3-day incubation (using different colors). This way, the changes would be visible to the reader.

Lines 162-164: What does PL mean?

Annotated reviews are not available for download in order to protect the identity of reviewers who chose to remain anonymous.

Reviewer 2 ·

Basic reporting

This manuscript investigated the potential of midgut gland homogenates of the sea urchin (S. nudus), starfish (P. pectinifera) and scallop (M. yessoensis) to biodegrade the plastic polyethylene (PE). Small pieces of PE were incubated with midgut gland homogenates for 3 days. Afterwards, Fourier-transform infrared (FTIR) spectra were taken of the plastics and compared to a control.

The topic of the study is very interesting and fits well into the current research in the field of polymer degradation. While the idea of the experiment is great, the experimental effort is kept to a minimum. With the goal to evaluate the enzymatic biodegradation, not enough was done to actually measure enzymatic activities in the samples to prove the observed degradation to be caused by enzymes. Furthermore, it would have been beneficial for this study to include more degradation parameters than only FTIR, since it is hard to set the findings into perspective. I see this manuscript in a revised form rather in a Short Communication letter than a full Research Article. I would also recommend to have this manuscript proofread/revised by a fluent speaker, to ensure that the readers can clearly understand the manuscript.
The introduction needs more background information and a lot more streamlining. It is for the most part unclear if this study is about fragmentation of plastics by enzymes or about bioremediation, since information are added paragraph after paragraph without following a common thread. The relevance of the chosen material (PE) and organisms needs to be emphasized more.

The results of this study show significant differences between the incubated and the control plastics, which can be an impactful basis for further research. However, since I am not a professional in FTIR analyses, I can only evaluate the statistical effort. There are clearly some information missing about what tests exactly were performed to obtain the p-values. Having a look in the supplementary data, I either missed the raw data or the authors only uploaded the already calculated ratios of the spectra. The rest of the experimental part is clearly described with detailed information.

In the discussion, the changes of the IR spectra in comparison with results from other studies is well written and plausibly discussed. However, the discussion fails to set the results in a broader context. This might due to the low amount of data generated by this study in a generally novel field of research where not many comparable studies are available, but also because of misinterpretation/wrong citation of references. The authors make strong assumptions about the ecological impact of the findings, without considering or discussing physiological factors. The discussion would benefit from focusing more on what the results of this study actually show and what can be interpreted from them.

The Conclusions section is missing, but the last paragraph of the discussion seems to be a conclusion. However, this conclusion is highly hypothetical and not based on the results of this study.

The Acknowledgments are missing.

Experimental design

no comment

Validity of the findings

no comment

Additional comments

I would not recommend to publish this article in the current state. However, I think with some effort and a thorough revision, the results of this study can be publishable. I provided some more specific comments below, that could help you to improve your manuscript.

Specific Comments:

Introduction
Line 40: Please rephrase „is very acute and requires a quick solution”, f.e. into “is a pressing issue for which a solution is urgently needed.”

Line 40-43: Please add more context. You are jumping from the problem of plastic pollution directly to plastic fragmentation. Maybe add more details on why plastic pollution is such a problem. What does it mean if plastic gets fragmented? What does fragmentation have to do with plastics in the environment or how does it affect the pollution problem?

Line 49: Which polychaete? Please specify.

Line 50: The abbreviation “MPs” is out of place. You previously write about microplastics, that is where this abbreviation should be placed.

Line 52: Remove “for example”. Rephrase into something like: “Labeled MPs from holothuria pseudofeces were resuspended in much higher numbers than MPs from bottom sediment”.

Line 54-63: This paragraph does sound very vague. “Presumably due to”, “A weak correlation”, “authors do not exclude the possibility that”. Either remove this paragraph or rephrase it.

Line 73: Annelids and echinoderms

Line 87-88: “may be” or “is”?

Line 84-92: For the reader it might be unclear, why the midgut gland homogenate of marine invertebrates was used. Is it because these organisms are especially exposed to (micro)plastics in their environment? If this is about plastic valorization and bioremediation why not use enzymes of bacteria or fungi which are way easier in handling and culturing? In general this introduction needs a bit more explanation of what exactly the scope of this study was and what the authors wanted to show.

Materials and Methods
Line 94: “Site of bivalves collection and material”. What about the collection of the other animals (starfish, sea urchin)?

Line 97: Why was a pH of 7.5 used? Does this pH correspond to the natural pH occurring in the digestive system of the used animals?

Line 98-99: Please specify the origins of the plastic used in this study, its brand name and from which producer/manufacturer it was bought.

Line 99: FTIR – please spell out a term the first time it is used.

Line 101-105: Why was the homogenate changed? How was the homogenate used: homogenate of only one specimen per plastic piece or homogenate of different specimen per plastic piece or were the homogenates pooled in advance? How many replicas were used?

Line 106-107: Why were the plastic pieces washed with water and ethanol? Please specify.

Line 108-109: Were the control samples also constantly agitated with magnetic stirring?

Line 137: Which base statistics do you mean? Please specify. How did you compare the ratios of the different indices (parametric or non-parametric t-test?) and how did you cope with the zero value of the control for CI and HI in your test? Did you check for homogeneity of variances?

Results
Where are the FTIR results of the chemical composition of the plastics, as described in section 2.1?
Line 144-145: “after 3 days” implies there were also other times measured. I assume there were no other time points?

Line 144: Significant changes are usually accompanied with a statistical value, here it would help to give p- or comparable values in the text.

Line 152-157: This paragraph should be moved into the discussion section, since it is not describing, but already discussing the results.

Line 159-161: Same as above, move to Discussion section.

Line 165-166: “Significant increase”. Please provide some p-values or comparable in the text, or add it to the table with the indice ratios, with a reference to the table in the text.

Line 167-168: Move sentence to discussion.

Line 172: What is “damage” indices calculation? The number of the table is missing.
Discussion

Line 179-180: “to some extent indicates the activation of the initial stage of polymer degradation”. Please explain further, it is still unclear for readers without the proper background to understand what you mean. What is the initial stage of polymer degradation? Why do oxygen-containing functional groups indicate degradation and to what extent? Here it would have been interesting to set the degree of degradation in relation with other parameters, f.e. surface SEM analysis of the plastic pieces.

Line 187-194: There are also natural polymers of animal origin, proteins such as collagen or elastin, but also polysaccharides such as chitin. Enzymes that hydrolyze these polymers are common in a wide variety of carnivorous/omnivorous marine invertebrates. Why are they not capable of hydrolyzing the polymer PE?
Maybe it would be a helpful addition to measure the enzyme activities of key polysaccharide degrading enzymes or in general the enzymatic activities in your samples.

Line 193 and 196: The author’s name of this study is “Trestrail”

Line 192-194: I am critical about the statement that this study suggests a role of cellulase in the degradation of PS due to significant induction of cellulase activity. Rather, the study finds no significant induction of cellulase with PE, which is the polymer used in your study. It also states: “PE and PS molecules bear little resemblance to cellulose, so it is difficult to see how cellulase would act on these polymers.” Why is this not discussed? This reference is as it is not supporting your discussion.

Line 195-196: What is a “long time”? Long in comparison to the retention time of food or other particles? I would rather stick to the actual time (e.g. for 10h). Please check this reference, since in this study there is nothing written about residence time of MPs in mussels!

Line 200: What is the point of this reference? Polymeric starch is a biodegradable polymer and structurally and degradation-wise very different from polyethylene. Furthermore, an induction of enzyme activity does not necessarily mean that these enzymes are involved in degradation. An increase in activity could occur due to molecular or sensory resemblance of the polymer or leaching additives to natural food.

Line 201-210: What do these significant chemical changes mean for biodegradation? “Relatively short-term exposure” is indeed relative, since three days is a “relatively long time” for indigestible particles to reside in the digestive system of marine invertebrates. Many marine organisms have mechanisms to cope with indigestible particles, and get rid of them through ejection (mussels), regurgitation (shrimp) etc. Therefore, particles will unlikely be exposed to the digestive enzymes for 3 days.

Line 228-235: Please specify about which polymers you are talking about, synthetic or natural polymers? Microorganisms play a key role in biodegradation of plastics, but the mechanisms of enzymatic degradation by microorganisms in a biofilm on the surface of a plastic differs a lot from the digestion mechanisms in an organism, especially considering the time factor. Although microorganisms feed on plastics as carbon source (when no other carbon source is available), why should invertebrates feed on synthetic polymers that have no nutritional value but carbon when other food is available? This discussion would benefit from adding some more information about the physiology of the animals used as test organisms and the likelihood of plastics being ingested and retained in the digestive system.

Line 244-249: Is this paragraph the conclusion of the study? Then please insert a heading. Based on the results this study provides, it is very far-fetched to make assumptions as stated in this paragraph. The authors fail to set the degree of biodegradation by digestive enzymes into perspective, yet assume a significant acceleration of the biotransformation of plastics in the environment by this process. Furthermore, the assumption that invertebrates participate in the removal of plastics from the environment needs to consider a lot more factors than just the enzymatic degradability of these compounds by digestive enzymes, such as interaction, ingestion and retention of plastics with biota, and last but not least the toxicity of these compounds on biota.

I understand that some of these aspects go beyond the scope of this study, but the study needs at least to address them in the discussion before making such concluding assumptions. I would recommend to focus the conclusion more on your results, making less hypothetical statements and if you do, please use more neutral phrases such as “this could indicate”, “the results might suggest” or “from a bigger perspective, this could mean”.

---

## Round 0.2 · Minor Revisions

Dear Dr. Mazur,

Your revised manuscript has been evaluated. However, as you see, the Reviewer has made some critical comments about statistical analysis and specified some uncertainties that must be addressed before the article is accepted. In addition, I have made several suggestions for revising the text.

When submitting the revised version of your manuscript, please state in your cover letter point-by-point which changes you have made in response to the reviews and where and why you have refused to follow a particular suggestion. The accordance between the changes and the reviewers' requests should be sufficiently transparent.

My suggestions:
“Digestive gland” is correct.
Patronymics are not need.
Plastic “fragments” are more suitable.
The authos were confused with “Holothuria”. The reviewer noticed that in the article “holothurian is written in lower case”. However, Holothuria is a genus name, but a the common name is sea cucumber. I suggest to change the phrase: “It was also observed that the ingestion and excretion of MPs by sea cucumber the sea cucumber Holothuria tubulosa increased its their bioavailability, although it they did not change the size. ForLabeled MPs from Holothuria pseudofeces of H. tubulosa were resuspended in much greater higher numbers than MPs from bottom sediment”.
Only three invertebrates have been studied – the scallop, the sea urchin and the sea star, so the conclusion about the important role of all typed of “multicellular organisms” in plastic cycle is hasty. It is more reliable to draw conclusion for a number of (or some) marine invertebrates.
“Moreover, such particles have been recorded in representatives of various taxa (more than a thousand species) of different trophic levels”. It should be clarified. Apparently, it is a matter of marine invertebrates.
“Higher eukaryotes” is too general a concept. In your case, the term “marine invertebrates” is more appropriate.
The aim of the study should be formulated at the end of the Introduction. In current version, the last sentence about the "trophic mode" of the studied species does not fit (does not agree) in the Introduction of the article.
Moreover, the authors confuse the feeding modes (filter-feeders, suspension feeders, grazers, etc.) and feeding type (carnivorous, herbivorous or omnivorous animal). “Filtrator” should be changed on a common “filter feeder”. Strongylocentrotus nudus is herbivores, “P. pectinifera is a carnivorous predator”. This is a repetition, sometimes carnivores are called predators.
In Experiment section, specify the type of plastic and the sizes of the pieces of plastic.

The last paragraph of the Conclusion is a speculation, since “symbiont microorganisms” have not been studied by the authors. I recommend briefly limiting the assumption about the possible role of microorganisms habiting the digestive tract.

In general, although the authors have made corrections, the fragmentary paragraphs and a lack of logical order are mostly observed. To improve the article, it is necessary to carefully evaluate the logical relationship between sentences in paragraphs, as well as the relationship between paragraphs. Keep in mind that a paragraph should not consist of a single sentence.

Reviewer 2 ·

Basic reporting

The authors have addressed the comments of the previous review round accordingly and provided a refined manuscript, which is greatly improved. There are only some minor things to address, which I commented below.

Specific Comments:

Introduction
Line 40-43: I would refrain from using the term “polymeric fragments”, since there are also natural polymers in the environment. Rather specify them as synthetic polymeric fragments, or better use the term microplastics.

Line 40-43: Please rephrase this paragraph into something like:
“A multitude of studies shows the global distribution of microplastic particles, that can be found in all environments (water, land, air). Microplastics can be taken up by a variety of organisms of different taxa and trophic levels, which has been reported by several studies. Microplastics (or the ingestion of microplastics) can have harmful effects on biota, f.e. (insert 2 or 3 examples).”

Line 144: My previous comment was not meant to reduce the impact from the results by removing the term “significant”. What I meant is, that if there was a significant effect, it is important to point this out and support this statement by adding the calculated p-value of the statistical test. Same for Line 165-166.

Discussion:
Line 179: I would recommend to add the paragraph prepared in the response document into the discussion, since it would help the readers to understand the impact of your analyses and the underlying mechanisms of biodegradation of plastic polymers.

Line 192-194: I maybe understand what you were trying to say, but the citation is still wrong. This statement was not made by Trestrail, but rather from other sources (Song et al., 2020; Jang et al., 2021). Therefore, you have to cite these sources and not the paper from Trestrail, since your citation indicates that this statement is a conclusion from the study of Trestrail, which it is not.

The last paragraph of the discussion appears already earlier in the text (around Line 216). Please delete.

Table 1:
It would be helpful to add the p-values to this table, where significant differences are indicated.

Experimental design

-

Validity of the findings

-

Additional comments

-

---

## Round 0.3 · Minor Revisions

Dear Dr. Mazur,
In the revised version, the authors took into account the most of the comments of the previous review and provided an improved manuscript. There are only some minor points to address, which I have commented below.

There was a misunderstanding. It was not possible to visualize the crossed out words in the sentence, and you inserted a mixture of the old and new versions. Please, cheak carefuly and correct as: “It was also observed that the ingestion and excretion of MPs by the sea cucumber Holothuria tubulosa increased their bioavailability, although they did not change the size. Labeled MPs from pseudofeces of H. tubulosa were resuspended in much greater numbers than MPs from bottom sediment”.

I wrote that “carnivorous predator” is a repetition, and sometimes carnivores are called predators, but you left the previous version “P. pectinifera is a carnivorous predator”. You should select either carnivores or predator here and in Discussion.

The word "also" appears 14 times in the text. This should be avoided.

---

## Round 0.4 · accepted · Accept

In the revised version the authors took into consideration all comments and remarks. I recommend accepting the manuscript for publication in PeerJ.